# Studying a Multi-Stage Flash Brine Recirculation (MSF-BR) System Based on Energy, Exergy and Exergoeconomic Analysis

Faezeh Farhadi, Mahdi Deymi-Dashtebayaz * and Edris Tayyeban

Center of Computational Energy, Department of Mechanical Engineering, Hakim Sabzevari University, Sabzevar 9617976478, Iran
* Correspondence: mahdi.deymi@gmail.com; Tel.: +98-514-401-2814

**Abstract:** Due to the lack of natural water resources and high consumption of water in industries, desalination systems are good options to supply water demands, especially in regions with a water crisis. If these wastes are used in thermal desalination cycles, in addition to improving efficiency and reducing energy consumption, the production of environmental pollutants can also be reduced. In this paper, the multi-stage flash brine recirculation (MSF-BR) system of the Abadan refinery is investigated from energy-exergy-exergoeconomic viewpoints. In addition, the effects of top brine temperature (TBT), number of stages and ambient temperature on the performance of the system are evaluated. The results at maximum brine temperature show that with increasing the TBT, the exergy efficiency, gained output ratio (GOR) and distillate water production increase by 34%, 47% and 47%, respectively. It is also found that if the number of stages in the heat rejection section increases to more than six stages, GOR will decrease. The exergoeconomic analysis results reveal that the relative cost difference increases by 94% with an increase in the number of stages. Finally, it is concluded that by using the waste heat of a refinery complex for heating steam to run the desalination system, there is a 9103 \$/year cost savings due to energy consumption reduction and $193 \times 10^4$ \$/year cost savings due to $CO_2$ emission reduction.

**Keywords:** multi-stage flash desalination; stage number; energy efficiency; exergy destruction; exergoeconomic

## 1. Introduction

Climate change and world population growth have led to an increase in water demand and consequently a decrease in water resources [1]. Around 40% of the world's population suffers from a lack of potable water which is expected to be increased in the future [2]. Although 70% of the Earth's surface is covered with water, only 3% of it is drinking water [3]. Desalination of seawater is one of the growing methods to supply potable water around the world [4]. Generally, the desalination methods are divided into thermal and electrical techniques. Thermal desalination systems are generally divided into two main methods, MSF (multi-stage flash) and MED (multi-effect desalination). The water production process in thermal systems uses evaporation and distillation processes. Therefore, the produced water is of good quality. Power water desalination systems, the most common of which is the RO (reverse osmosis) method, use a membrane to filter and purify water. Moreover, electrodialysis (ED) and nanofiltration (NF) are other desalination methods using a membrane. While employing thermal desalination methods leads to pollutant emissions due to using fossil fuels, the waste heat of industrial units can be utilized as a heat source for thermal desalination systems. As shown in Figure 1, after reverse osmosis (RO), thermal desalination technologies account for a significant share of total desalination methods in the world. Therefore, optimization of such thermal desalination systems has considerable impact on reduction of fuel consumption and pollutant emissions. Multi-stage flash (MSF) is one of the most used thermal desalination techniques [5]. However, MSF desalination

systems consume a large number of fossil fuels which leads to an increase in air pollution and a decrease in non-renewable energy sources. As a result, energy consumption optimization of thermal desalination units like MSF can reduce energy consumption and water production cost [6].

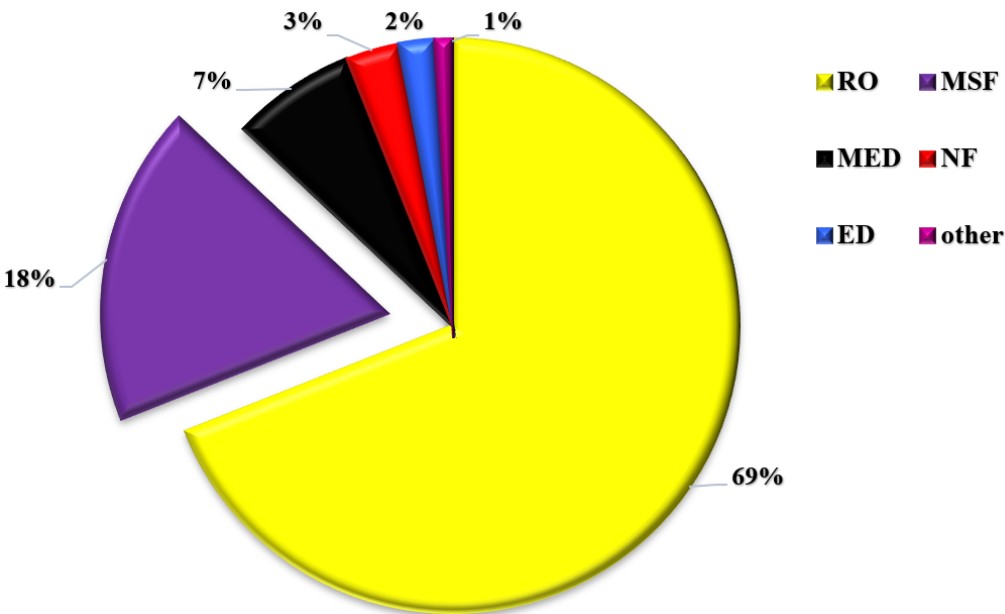

**Figure 1.** Share of different desalination facilities by technology [5].

The thermal desalination systems are categorized into two main groups—multi-flash and multi-effect desalination systems. The multi-stage flash desalination system requires low-pressure heating steam to run and medium-pressure steam to create a vacuum in the system. An MSF desalination system with 4 to 40 stages can produce 1000 to 35,000 m³/day of potable water [7]. Generally, in this system, the seawater or brackish water, which is called feed brine water, enters the last stage being pre-heated in each stage and moves towards the first stage. Then, it goes to the brine heater and reaches a higher temperature and pressure through heat transfer with incoming heating steam. After that, it flows back to the stages that have ever lower pressure and temperature. This causes a small fraction of it to evaporate partially (flash) in each stage which is then condensed after contacting the tubes containing the feed brine water and produces potable water. The required 90–110 °C thermal energy for the brine heater of the MSF system can be supplied through burning fossil fuels, the waste heat of refinery or power plants, solar energy, etc. [8]. Due to the high share of MSF desalination among thermal desalination systems, numerous studies have been conducted on the optimization and performance improvement of MSF systems. The replacement of non-renewable with renewable energies and waste heat of energy systems is another field of study in this area which reduces the emissions, costs and increases efficiency [9,10]. In addition, as known, the outlet brine of MSF systems is still warm. One method is to reuse the thermal energy of outlet brine to increase the temperature of inlet feed brine. There are some studies that worked particularly on performance of multi-stage flash with brine recycle (MSF-BR) systems.

Al-Weshahi et al. [11] analyzed a 3800 m³/h MSF-BR desalination system with 16 stages for heat recovery and three stages for heat rejection using IPSEpro software. Results show that the maximum and minimum exergy destructions are related to heat recovery and pump, respectively. Furthermore, they found that by increasing the number of stages to eight stages, the exergy efficiency increases by 8% while it drops by 4% when the number of stages exceeds eight stages. Ben Ali and Kairouani [12] optimized a 10-stage MSF-BR desalination system with a capacity of 26,700 m³/day comprised of ten 16-stage units by means of MATLAB programming. The multi-objective optimization was aimed

at maximizing potable water production, minimizing the heating steam flow for minimum thermal energy consumption and minimizing the pump flow for minimum power consumption. It was concluded that with increasing the heating steam flow, the potable water production increases while it decreases with increasing the seawater temperature. Al Ghamdi and Mustafa [13] developed a MATLAB code to model the performance of an MSF desalination system located in Saudi Arabia with a water production capacity of 159 kg/s in the winter and 200 kg/s in the summer. Results obtained for exergy destruction and exergy rate show that by exergy efficiency of 74.9%, the maximum exergy destruction occurs in the heat recovery section and decreases down to 69.2% with increasing the number of stages from 25 to 31 stages. In addition, the second law of thermodynamic efficiency of the system in winter and summer was obtained at 3.22% and 2.84%, respectively. They adopted some approaches such as Once Through (MSF-OT) and Brine Mixing (MSF-M) to minimize the exergy destructions in the system. Sanaye and Asgari [14] performed multi-objective optimization over a combined cycle power plant coupled with an MSF desalination system based on 4E (energy, exergy, economic and environmental) analysis using Thermo-flow software. They considered investment, operational and $NO_x$ emissions penalty costs as well as total exergy destruction of cycle as objective functions. Additionally, the impacts of gas turbine partial load, ambient temperature and fuel cost changes on optimal values of the drum pressure of HRSG, pinch point temperature of heat recovery steam generator (HRSG), top brine temperature of MSF, last stage temperature of MSF and number of MSF stages as design parameters were investigated. Their results showed that with increasing the ambient temperature, the amount of potable water production, steam turbine power generation, exergy destruction, fuel consumption and emissions penalty costs decrease while the payback period increases due to a decrease in annual income. It was also revealed that by increasing the last stage and top brine temperature of the MSF system, the temperature difference between successive stages of heat recovery and heat rejection sections increases which leads to a decrease in the heat transfer area, exergy destruction and costs. Bandi et al. [15] modeled the once-through MSF (MSF-OT), brine-mixing MSF (MSF-M) and brine recycle MSF (MSF-BR) desalination configurations mathematically to find the optimal solutions for decision variables taking into account the production cost of potable water as an objective function. They reported significant improvements in important parameters of the MSF desalination system.

Mohamed Al-Hamahmy et al. [16] presented a novel method by extracting the cooling brine and re-injecting it to stages without passing through the brine heater. This will improve the thermodynamic and economic efficiency of the MSF desalination system by reducing the heat transfer area and power consumption. The performance of the system using single-point brine extraction was better than multiple-point brine extraction and a 7.23% increase in gained output ratio (GOR), 3.47% decrease in power consumption and 3.90% decrease in total cost were achieved. Sharaf Eldean and Soliman [17] utilized the waste gas of oil refineries to run the hybrid thermal desalination processes based on three scenarios including MSF-BR+MED (multi-effect desalination), MSF-BR+MED+ORC (organic Rankine cycle) and MSF-BR+GTC (gas turbine cycle). Final results showed that by using 5 $m^3$/h of waste gases, the third scenario can produce 100 $m^3$/day of freshwater and 60 MW of power with zero UHC while the amount of freshwater production in the second scenario is 38,000 $m^3$/day. In addition, the value of exergy efficiency for first, second and third scenario was calculated equal to 62.73%, 23% and 23%, respectively. It was also concluded that the first scenario has no cost advantage, the second scenario is good for high freshwater production and the third scenario is applicable when there are huge amounts of waste gases. Carrasquer et al. [18] estimated the unit exergy cost of MED, MSF, RO and ED (electro dialysis) desalination techniques by combining the exergy cost analysis with Transfer Function Analysis based on recovery ratio, energy requirements and salts concentrations, plant capacity and organic matter recovery. They found that the unit exergy cost of membrane-based techniques (RO and ED) varies from 2 to 7 while it ranges from 10 to 26 for thermal-based techniques (MED and MSF). The exergy efficiency of RO

was obtained at 47% which is much higher than the exergy efficiency of thermal-based techniques (3–10%). There is also extensive research that exclusively worked on MSF desalination systems from different viewpoints. A summary of important studies about MSF systems is presented in Table 1.

**Table 1.** Summary of studies conducted on MSF systems.

| Reference | Technology | Analysis | Capacity | Result |
|---|---|---|---|---|
| Nasser and Mabrouk [19] | MSF-BR and MSF-OT | Exergy and energy | 30.07 MIGD | Heat transfer area and pumping power of MSF-OT are lower than MSF-BR. |
| Al-Othman et al. [20] | MSF | - | 1880 m$^3$/day | Two parabolic trough collectors can supply 76% of energy requirements. |
| Nafey et al. [21] | MSF-BR | Economic, exergy and energy | 18,000 m$^3$/day | The unit product cost increases by 21% by decreasing the load down to 50% of the design point. |
| Hanshik et al. [22] | MSF-OT | Energy and exergy | - | The performance of MSF plants improves with increasing the top brine temperature. |
| Fiorini and Sciubba [23] | MSF | Exergy, energy and economic | 347.07 kg/s | The cogeneration MSF plant with a high top brine temperature and a low number of stages is preferred. |
| Yagnaseni Roy et al. [24] | MSF-OT | Energy balance | 3384 kg/s | The performance ratio increases by 41.5% with increasing the top brine temperature to 161 °C. |
| Hasan Baig et al. [25] | MSF | Energy balance | 378.8 kg/s | High temperature difference in the first stages and decreasing it in the last stages leads to a 3.13% decrease in condenser area and a 2.1% increase in performance ratio. |

Iran is one of those countries that is experiencing a serious water crisis. Industries such as chemicals, food and metals account for a large part of water consumption. Therefore, the use of desalination plants is a good option to tackle the water shortage and supply the water demand. Furthermore, the waste heat of petroleum refineries located in coastal areas can be utilized for the thermal desalination of seawater. This study investigates the multi-stage flash brine recirculation (MSF-BR) system of the Abadan refinery which has 18 stages in the heat recovery section and four stages in the heat rejection section.

In the first stage, the current research has investigated and modeled the water softener system installed in the refinery. The system in the refinery uses natural gas to supply the energy required for the desalination system. However, according to field investigations, heat losses in the steam production sector are significant. Therefore, in this research, the feasibility of using existing steam waste with the aim of supplying the energy needed for the desalination system is proposed. The potential of employing waste heat of a refinery complex as the heating steam of a desalination unit is studied based on energy, exergy and exergoeconomic analysis. The main goal of choosing energy, exergy and exergy analysis is to achieve a logical optimization by considering important aspects. Exergoeconomics is the branch of thermodynamics that combines exergy and economic analysis to provide the system designer with information not available through conventional energy analysis and economic evaluations. The results are firstly validated using the existing data of the Abadan refinery and then, the effects of some operational parameters on the performance of the system are examined. Finally, the annual amounts of cost savings due to energy consumption reduction and emissions reduction are calculated.

## 2. System Description

The multi-stage flash brine recirculation (MSF-BR) system investigated in this study is located in the Abadan refinery complex which is used to desalinate water from the Arvand River for drinking and steam boilers. Currently, the required heating steam for

the brine heater of the desalination system is supplied by existing steam boilers of the refinery complex. Figure 2 shows a schematic view of the MSF-BR desalination system. It is composed of 18 stages in the heat recovery section and four stages in the heat rejection section. The seawater (brine) enters the last stage of the rejection section through tubes (point E), is pre-heated and flows towards the recovery section. At this point (point F), some part of the seawater is returned to the sea and the rest of it is mixed with recycled brine (point C) and enters the recovery section and goes to the first stage. Then, it is pre-heated again in each stage using latent heat of the distillate vapor and moves to the brine heater and is heated by heating steam to reach a higher temperature and pressure than in the first stage. Now, it flows back at a temperature of 115 °C under saturated conditions to the first stage which is a vacuum (point A). This causes a fraction of inlet brine to evaporate partially (flash) which is then condensed through heat transfer with tubes of feed brine and is drained. The rest of the inlet brine flows to the next stage and this process continues until the last stage and the condensed water is collected as a distillate water product (point B). Some of the remaining concentrated brine is discharged (point D) and the rest is recycled at a temperature of 42 °C. The existing desalination plant of the Abadan refinery complex is shown in Figure 3.

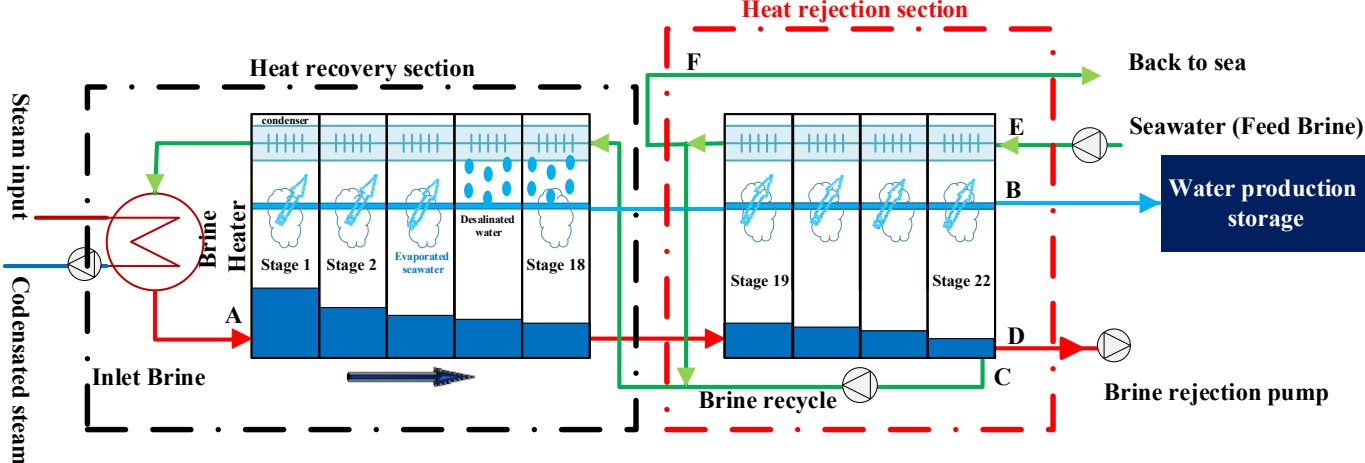

**Figure 2.** Schematic view of a MSF-BR unit.

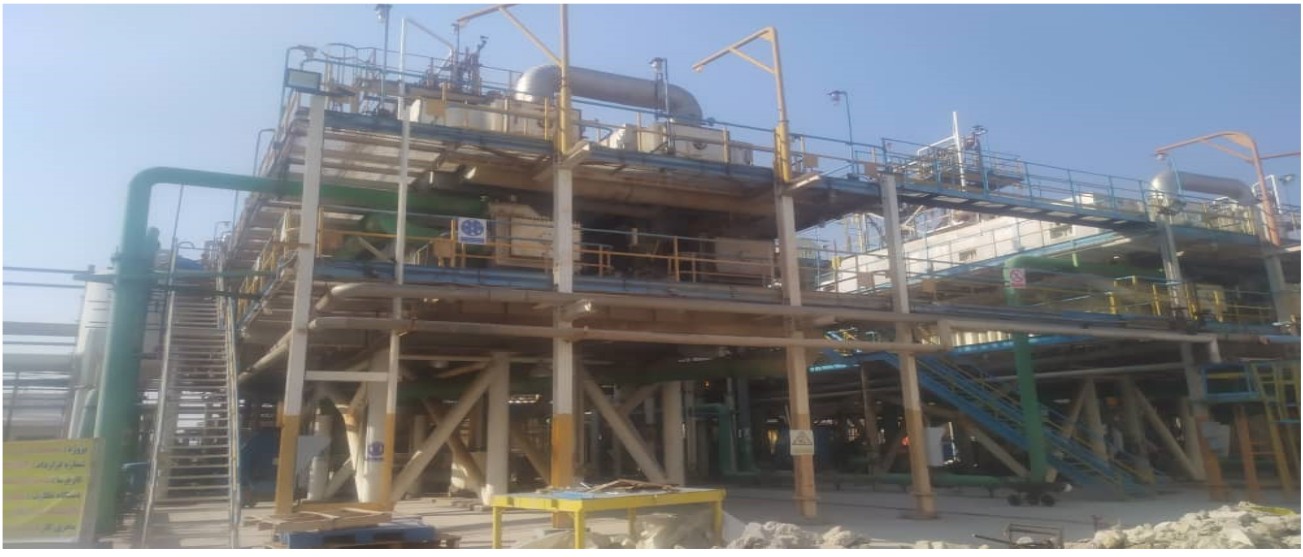

**Figure 3.** MSF-BR desalination system of the Abadan refinery complex.

## 3. Modeling

The modeling of the MSF-BR desalination system of the Abadan refinery is conducted based on energy, exergy and exergoeconomic analysis. Then, the effects of top brine temperature (TBT), number of stages and ambient temperature on energy efficiency and exergy efficiency of the desalination system are investigated. Table 2 summarizes the basic information used to model the desalination system based on the following assumptions:

- The latent heat of evaporation and condensation is assumed constant.
- The distillate water is salt-free.
- The variations in boiling temperature of brine at the recovery section outlet are small.
- Pressure and temperature are assumed to be constant in each stage.
- The heat capacity of brine is considered equal to that of distillate water at the same temperature.
- The heat transfer to surroundings is neglected.
- The pressure drop is negligible.

**Table 2.** Modeling parameters [26,27].

| System Parameter | Value |
|---|---|
| Pump isentropic efficiency $\eta_p$ (%) | 70 |
| Brine heater efficiency $\eta_{HEX}$ (%) | 90 |
| Temperature of vapor in the first stage (°C) | 111.2 |
| Pressure of vapor in the first stage (kPa) | 200 |
| Feed water salinity (ppm) | 8935 |
| Freshwater salinity (ppm) | 0 |
| Ambient pressure $P_o$ (bar) | 1.01 |
| Ambient temperature (°C) | 25 |

### 3.1. Energy Analysis

Taking into account the assumptions and conditions presented in the previous section, the energy analysis is conducted on the desalination system based on the energy conservation equation. Moreover, mass conservation is also applied to the concentration of water, steam and salt. The mass and energy conservation equations are generally written as:

$$\sum \dot{m}_{in} = \sum \dot{m}_{out} \tag{1}$$

$$\sum_{out} \dot{m} \times h - \sum_{in} \dot{m} \times h = \dot{Q}_{cv} - \dot{W}_{cv} \tag{2}$$

$$\sum \dot{m}_{in} \times X_{in} = \sum \dot{m}_{out} \times X_{out} \tag{3}$$

where $\dot{m}, h, \dot{Q}_{cv}, \dot{W}_{cv}$ and X are mass flow rate, specific enthalpy, heat transfer rate, work rate and concentration, respectively.

The inlet brine enters the first stage in a saturated condition after receiving the inlet energy from the system. The amount of energy received by inlet brine from heating steam in the brine heater is calculated using Equation (4):

$$Q_{in_s} = m_A C_{ps}(T_A - T_{sat_{steam}}) + m_A h_{fg} \tag{4}$$

Due to existence of impurities in brine water, Equation (5) is used to find the temperature for the range of 10 < T < 180 °C and 1 < TDS < 160,000 ppm [28].

$$T_i = T_{sat_i} + BPE \tag{5}$$

where BPE is an increase in the boiling point of inlet brine and is calculated using Equations (6)–(9).

$$BPE = X\left(A + BX + CX^2\right) \tag{6}$$

$$A = 8.325E - 2 + 1.883E - 4 \times T_i + 4.02E - 6 \times T_i^2 \tag{7}$$

$$B = -7.625E - 4 + 9.02E - 5 \times T_i - 5.2E - 7 \tag{8}$$

$$C = 1.522E - 4 - 4E - 6 \times T_i - 3E - 8 \times T_i^2 \tag{9}$$

In each stage, the distillate vapor is condensed through heat transfer with a tube of inlet brine. The required heat transfer area is computed by Equation (10).

$$A(i) = \frac{m_{s_1} h_{fgf}}{(U_i(T_1 - T_2))} \tag{10}$$

In the above equation, $U_i$ is the overall heat transfer coefficient for each stage and is calculated using the following equation [9]:

$$U(i) = 1.6175 + 1.537E - 4(T_i) - 1.825E - 4(T_i)^2 + 8.026E - 8(T_i)^3 \tag{11}$$

As described, part of the brine is desalinated and the remaining brine flows to the next stage with a temperature drop. Equation (12) is used to obtain the amount of temperature difference between successive stages where the temperature of the last stage is achieved from Equation (13).

$$T_{st.m} = T_{Ambient} + 10 \tag{12}$$

$$\Delta T = (T_1 - T_{st.m})/n \tag{13}$$

The gained output ratio (GOR) which is used to describe the performance of a thermal desalination system is defined as the ratio of distillate water to inlet heating steam [29]:

$$GOR = \frac{\left(\sum_{i=1}^{n} m_{d_i}\right)}{m_s} \tag{14}$$

where $m_{d_i}$ is the total amount of distillate water produced by the desalination system and $m_s$ is the amount of inlet heating steam to the brine heater.

*3.2. Exergy Analysis*

In addition to energy analysis, the maximum useful work performed during the desalination processes is obtained using exergy analysis based on the second law of thermodynamics. The inlet exergy to desalination system and exergy destruction of the first stage is calculated using Equations (15) and (16), respectively [30].

$$Ex_{in}(1) = \left(1 - \left(\frac{T_{swi}}{T_{si}}\right)\right)Q_{in} + m_{sw}[(h_{out,B} - h_{in,B}) - T_0(s_{out,B} - s_{in,B})] \tag{15}$$

$$Ex_{destruction} = m_s((h_{s1} - h_{sout}) - T_{sw_i}(S_{s1} - S_{sout})) - m_{sw}C_{p1}(T_{ww_o} - T_{sw_i}) \\ - T_{sw_i}\ln\left(\frac{T_{br_1}}{T_{in,B}}\right) - m_{d_1}h_{fg_1}(1 - (\frac{T_{sw_i}}{T_1})) \tag{16}$$

Finally, the exergy efficiency is the ratio of outlet exergy to inlet exergy as following equation [31]:

$$\eta_{ex} = \left(\frac{Ex_{in} - Ex_{destruction}}{Ex_{in}}\right) \tag{17}$$

The mass, energy and exergy balance equations used for each component of the desalination system are listed in Table 3.

**Table 3.** Mass, energy and exergy balance equations for desalination system components.

| Equation | Balance Equation | Component |
|---|---|---|
| Heat Exchanger | Mass balance<br>Energy balance | $\dot{m}_{in,v} + \dot{m}_{in,b} = \dot{m}_{out,b} + \dot{m}_{out,v}$<br>$Q = UA\Delta T_{LMTD}$<br>$\dot{m}_{in,v}[(h_{in} - h_0) - T_0(s_{in} - s_0)] + \dot{m}_{in,b}[(h_{in} - h_0) - T_0(s_{in} - s_0)]$<br>$= \dot{m}_{out,v}[(h_{out} - h_0) - T_0(s_{out} - s_0)]$<br>$+ \dot{m}_{out,b}[(h_{out} - h_0) - T_0(s_{out} - s_0)] + Ex_{destruction}$ |
| Pump | Mass balance<br>Energy balance | $\dot{m}_{in} = \dot{m}_{out}$<br>$W_{pump} = m_{ww}\nu_{ww}\left(\frac{P_2 - P_1}{\eta_{pump}}\right)$<br>$\dot{m}_{in}[(h_{in} - h_0) - T_0(s_{in} - s_0)] + W_{pump} =$<br>$\dot{m}_{out}[(h_{out} - h_0) - T_0(s_{out} - s_0)] + Ex_{destruction}$ |
| Stage 1 | Mass balance<br>Energy balance | $\dot{m}_{in,b} + \dot{m}_{in,d} + \dot{m}_{in,cw} = \dot{m}_{out,b} + \dot{m}_{out,d} + \dot{m}_{out,cw}$<br>$(\dot{m}_{in,b} \times h_{in,b} + \dot{m}_{in,d} \times h_{in,d} + \dot{m}_{in,cw} \times h_{in,cw}) - (\dot{m}_{out,b} \times h_{out,b} + \dot{m}_{out,d} \times h_{out,d} + \dot{m}_{out,cw} \times h_{out,cw})$<br>$\dot{m}_{in,b}[(h_{in} - h_0) - T_0(s_{in} - s_0)] + \dot{m}_{in,d}[(h_{in} - h_0) - T_0(s_{in} - s_0)]$<br>$+ \dot{m}_{in,cw}[(h_{in} - h_0) - T_0(s_{in} - s_0)]$<br>$= \dot{m}_{out,b}[(h_{out} - h_0) - T_0(s_{out} - s_0)]$<br>$+ \dot{m}_{out,d}[(h_{out} - h_0) - T_0(s_{out} - s_0)]$<br>$+ \dot{m}_{out,cw}[(h_{out} - h_0) - T_0(s_{out} - s_0)] + Ex_{destruction}$ |

*3.3. Exeregoeconomic Analysis*

Exergoeconomics which combines the exergy analysis, economic and environmental analysis, evaluates the system based on thermodynamic performance, economic principles and environmental considerations. Exergoeconomic analysis provides information about the costs associated with system components, exergy destruction and thermodynamic inefficiencies. In this analysis, the system is exergoeconomically examined by using the cost balance equation (Equation (18)). In this equation, the inlet exergy cost rate plus the total cost rate of each stage is equal to the outlet exergy cost rate [4].

$$\dot{C}_{in,Brine\ water} + \dot{C}_{in,Fresh\ water} + \dot{C}_{in,Sea\ water} + \dot{Z}_k \tag{18}$$
$$= \dot{C}_{out,Brine\ water} + \dot{C}_{out,Fresh\ water} + \dot{C}_{out,Sea\ water}$$

where $\dot{Z}_{tot,k}$ is calculated using Equation (19) and is the total cost rate representing the entire cost related to capital investment, operating and maintenance cost of the component.

$$\dot{Z}_{tot,k} = \dot{Z}_k^{CI} + \dot{Z}_k^{OM} \tag{19}$$

In the above equation, the capital investment cost rate ($\dot{Z}_k^{CI}$) and maintenance cost rate ($\dot{Z}_k^{OM}$) for each component are obtained according to Equations (20) and (22), respectively.

Additionally, the capital investment cost rate is computed using equations listed in Table 4 and Equation (20).

$$\dot{Z}_k^{CI} = \left(\frac{CRF}{\tau}\right)Z_k \tag{20}$$

**Table 4.** Capital investment cost ($Z_k$) for desalination system components.

| Component | Capital Investment Cost |
|---|---|
| Brine Heater | $130 \times \left(\frac{A_{BH}}{0.093}\right)^{0.78}$ |
| Pump | $705.48 \times (W_{pump}^{0.71}) \times \left(1 + \frac{0.2}{\eta_{pump}}\right)$ |
| Stage | $430 \times 0.582 \times U_{stage} \times A_{stage}$ |

In Equation (20), $\tau$ is the annual operating hours of the system which is assumed equal to 7500 h. Moreover, CRF is a capital recovery factor and is calculated using the following equation:

$$CRF = \frac{i \times (1+i)^n}{(1+i)^n - 1} \tag{21}$$

where i is the interest rate which is considered 0.1 and n is the component life cycle which is assumed to be 20 years.

$$\dot{Z}_k^{OM} = \gamma_k Z_k \tag{22}$$

The maintenance factor ($\gamma_k$) in the equation of maintenance cost rate (Equation (22)) is 1.06 [32].

The exergetic cost rate of inlet and outlet flow for each component are calculated using cost per exergy unit and exergy rate as following equations, respectively [33].

$$\dot{C}_{in} = c_{in} \times \dot{Ex}_{in} \tag{23}$$

$$\dot{C}_{out} = c_{out} \times \dot{Ex}_{out} \tag{24}$$

Equations (25)–(29) are used to find the cost per unit exergy of fuel, cost per unit exergy of product, cost rate of exergy destruction, exergoeconomic factor and relative cost difference which are the other exergoeconomic parameters in this study. Furthermore, the exergoeconomic balance equations for different components of the desalination system are summarized in Table 5.

$$C_f = \frac{\sum \dot{C}_{Fuel}}{\sum \dot{E}_{Fuel}} \tag{25}$$

$$C_p = \frac{\sum \dot{C}_{Product}}{\sum \dot{E}_{Product}} \tag{26}$$

$$\dot{C}_D = C_{Fuel}.\dot{E}_D \tag{27}$$

$$f_k = \frac{\dot{Z}_k}{\dot{Z}_k + \dot{C}_D} \tag{28}$$

$$r_k = \frac{C_P - C_F}{C_F} \tag{29}$$

**Table 5.** Exergoeconomic balance equations for desalination system components.

| Component | Exergoeconomic Equation | Auxiliary Equation |
|---|---|---|
| Brine Heater | $\dot{C}_{in,sea\ water} + \dot{C}_{in,vapor} + \dot{Z}_{Brine\ heater}$ $= \dot{C}_{out,sea\ water} + \dot{C}_{out,vapor}$ | $c_{in,v} = c_{out,v}$ |
| Pump | $\dot{C}_{in,water} + \dot{Z}_{Pump} = \dot{C}_{out,water}$ | $c_{b,recycle} = c_{in,cw18}$ |
| Stage | $\dot{C}_{in,Brine\ water} + \dot{C}_{in,Fresh\ water} + \dot{C}_{in,Sea\ water} + \dot{Z}_{Stage}$ $= \dot{C}_{out,Brine\ water} + \dot{C}_{out,Fresh\ water}$ $+ \dot{C}_{out,Sea\ water}$ | $c_{in,b} = c_{out,b}$ $c_{in,cw} = c_{out,cw}$ |

## 4. Validation

The results obtained in the current study for GOR, inlet mass flow rate of heating steam and mass flow rate of distillate water are compared with the existing data of the Abadan refinery desalination system for validation. As presented in Table 6, there is a good agreement between modeling results and real data.

**Table 6.** Comparison of modeling results and real data.

| Parameter | Current Study | Existing Data | Error |
|---|---|---|---|
| GOR | 6.405 | 6.5 | 1.46% |
| Inlet mass flow rate of feed brine | 371 (kg/s) | 356 (kg/s) | 4% |
| Mass flow rate of distillate water | 57 (kg/s) | 55.69 (kg/s) | 2.3% |

## 5. Results and Discussion

In this section, the results of energy, exergy and exergoeconomic analysis on the multistage flash brine recirculation (MSF-BR) system of the Abadan refinery are presented. In addition, the effects of some operational parameters on the performance of the system are investigated. Figure 4 shows the variations in pressure and temperature in each stage of the desalination system. As discussed, when the inlet brine enters the stage it is flashed into vapor due to pressure difference. Then, the distillate vapor is condensed through heat transfer with feed brine which causes the feed brine to be pre-heated. The remaining brine flows to the next stage which has lower pressure and the process repeats. Therefore, as is seen in Figure 4, the temperature and pressure of each stage are lower than those of the previous stage.

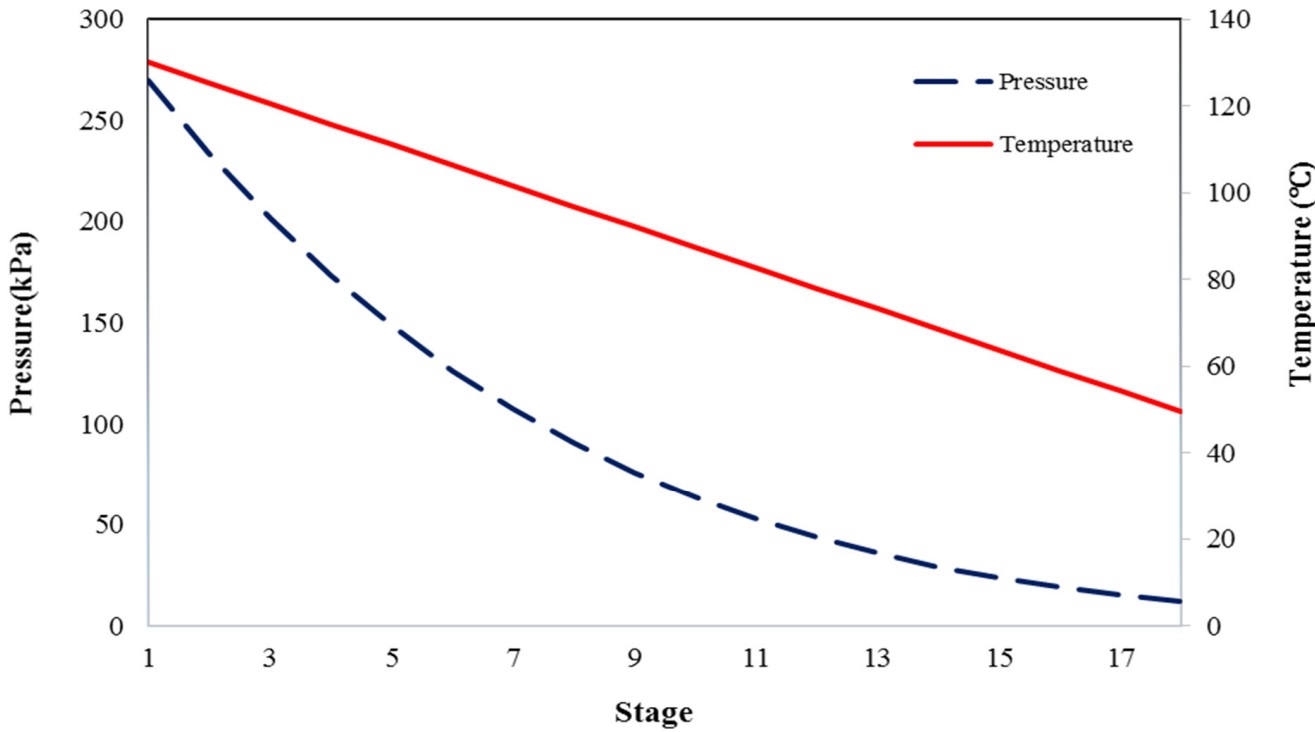

**Figure 4.** Variations of temperature and pressure in each stage.

The variations of total dissolved solids (TDS) and inlet energy in each stage of the desalination system are depicted in Figure 5. Since the pressure and temperature of inlet brine in each stage decrease, the inlet energy decreases from 194,505 kW in the first stage to 59,384 kW in the last stage. Furthermore, due to partial evaporation of inlet brine in each stage, the concentration of outlet brine increases from 31,448 ppm in the first stage to 47,353 ppm in the last stage.

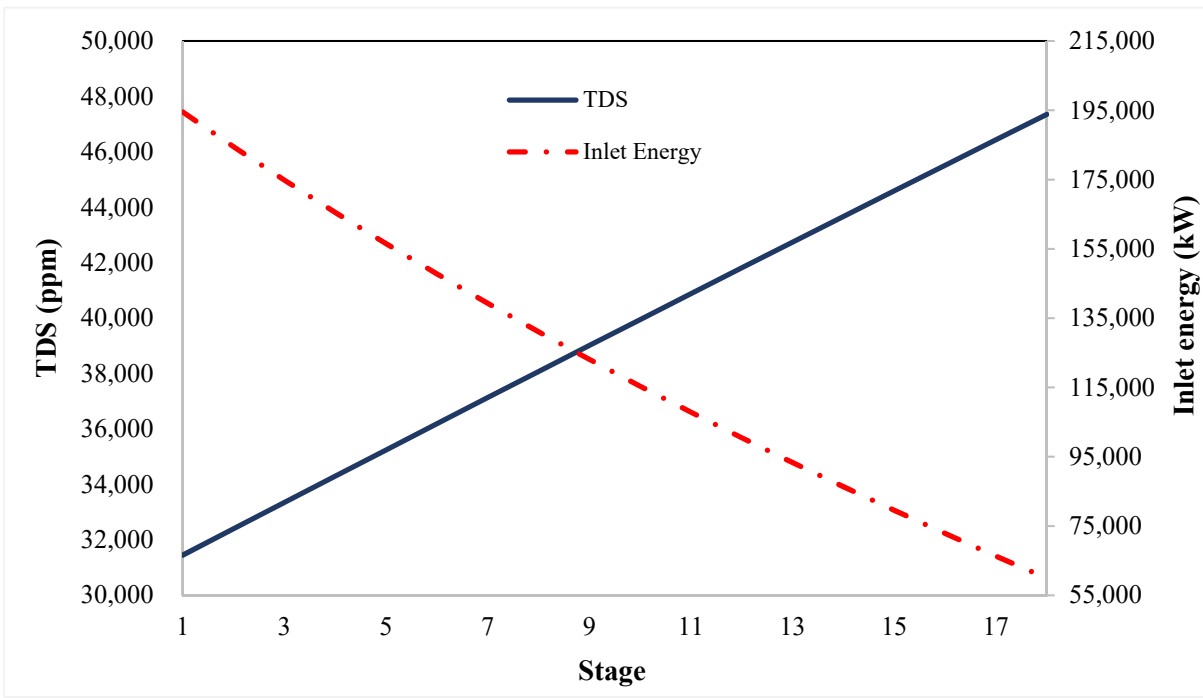

**Figure 5.** Variations of TDS and inlet energy in each stage.

Figure 6 demonstrates the effect of ambient temperature on produced distillate water in a desalination system with a different number of stages. With increasing the ambient temperature, the temperature of feed brine (seawater) also increases. Therefore, given a fixed heat transfer rate in the brine heater, the inlet brine enters the stages with higher temperature and energy which leads to higher evaporation and distillate water production. In addition, the amount of distillate water production increases by increasing the number of stages from 16 to 20 stages.

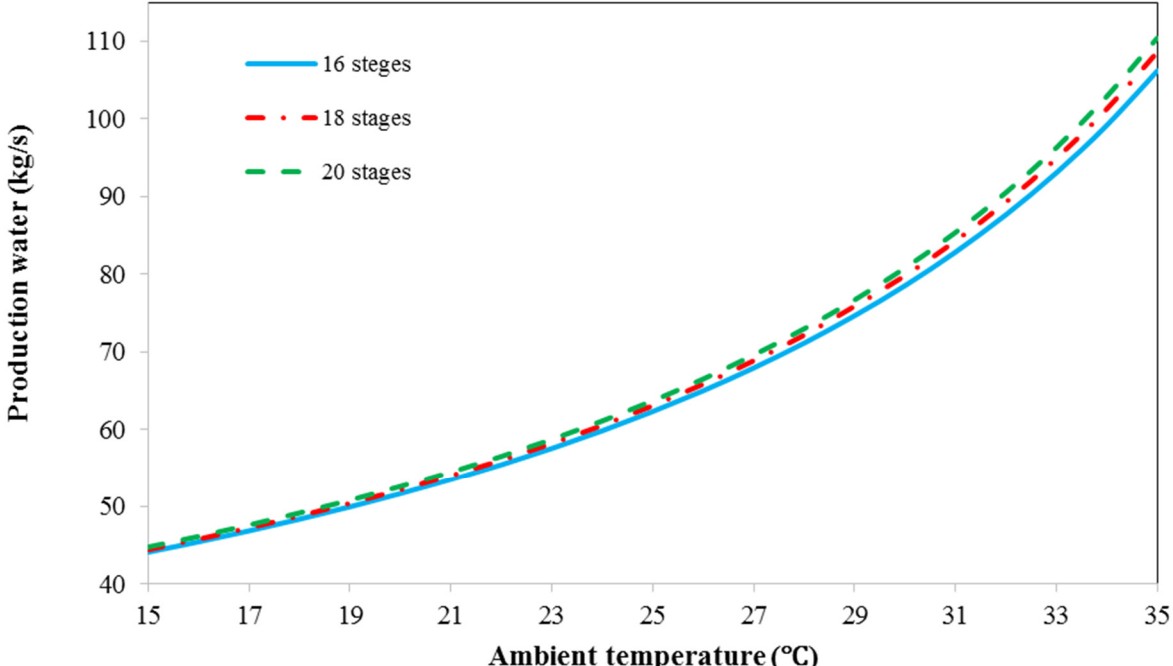

**Figure 6.** Variations of distillate water in terms of ambient temperature for the different number of stages.

### 5.1. Effect of TBT

The effect of top brine temperature (TBT) on produced distillate water and heat transfer area of the desalination system is shown in Figure 7. As it is clear, the increase in top brine temperature leads to higher inlet energy to stages and as a result, higher distillate water is produced. It is found that with increasing the TBT from 100 °C to 140 °C, distillate water production increases from 33.52 kg/s to 63.1 kg/s. Moreover, due to higher evaporation in each stage, a higher heat transfer area is needed. Results show that when TBT increases from 100 °C to 140 °C, the heat transfer area increases by 242 $m^2$.

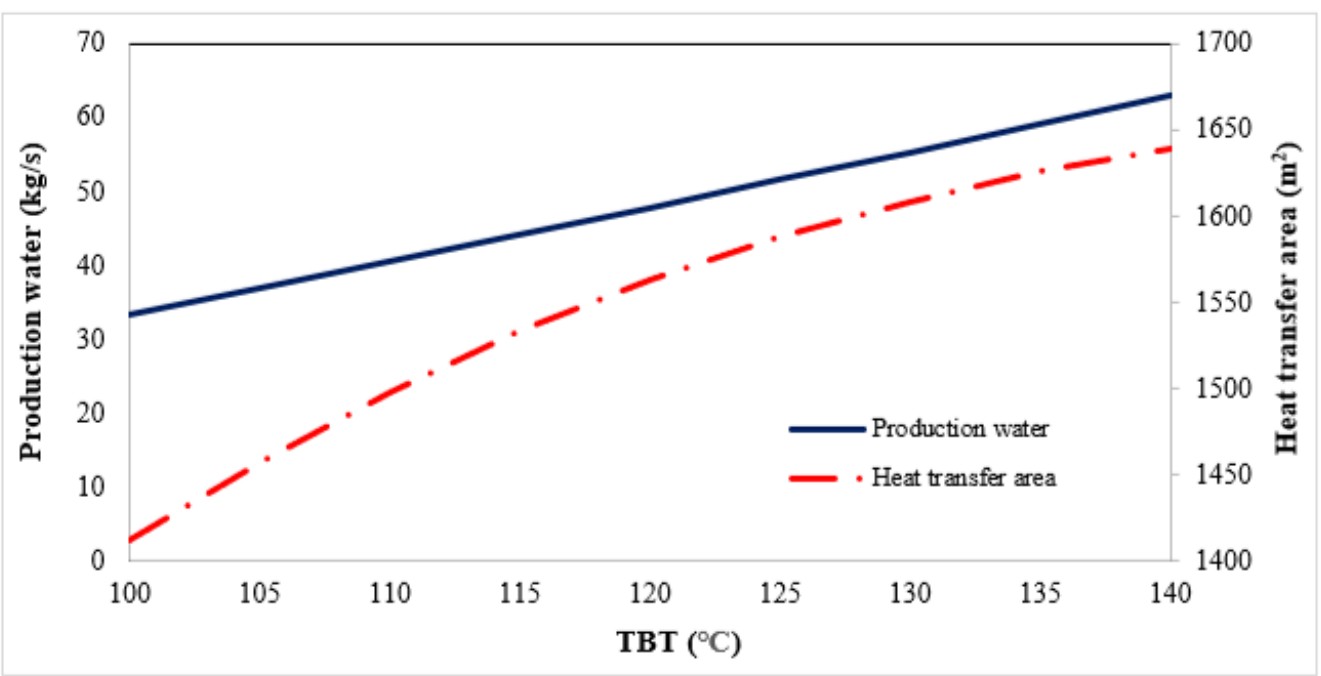

**Figure 7.** Variations of distillate water and heat transfer area in terms of top brine temperature.

The effect of top brine temperature (TBT) on gained output ratio (GOR) and exergy efficiency of the desalination system is shown in Figure 8. Assuming constant heating steam flow, with increasing the top brine temperature, the amount of distillate water increases which leads to an increase in GOR. It is seen that when TBT increases from 100 °C to 140 °C, GOR increases from 3.25 to 6.12. Furthermore, the exergy efficiency also increases from 47% to 63%.

### 5.2. Effect of Number of Stages

Figure 9A,B illustrate the effect of the number of stages on produced distillate water and heat transfer area of the heat recovery section and heat rejection section in the desalination system, respectively. With increasing the number of stages, a higher amount of inlet brine is evaporated, and therefore, higher distillate water is produced. In addition, due to higher heat transfer and the number of stages, a higher heat transfer area is required. As shown in Figure 9A,B, the increase in distillate water production and heat transfer area of the heat recovery section is higher than the water production and heat transfer area of the heat rejection section.

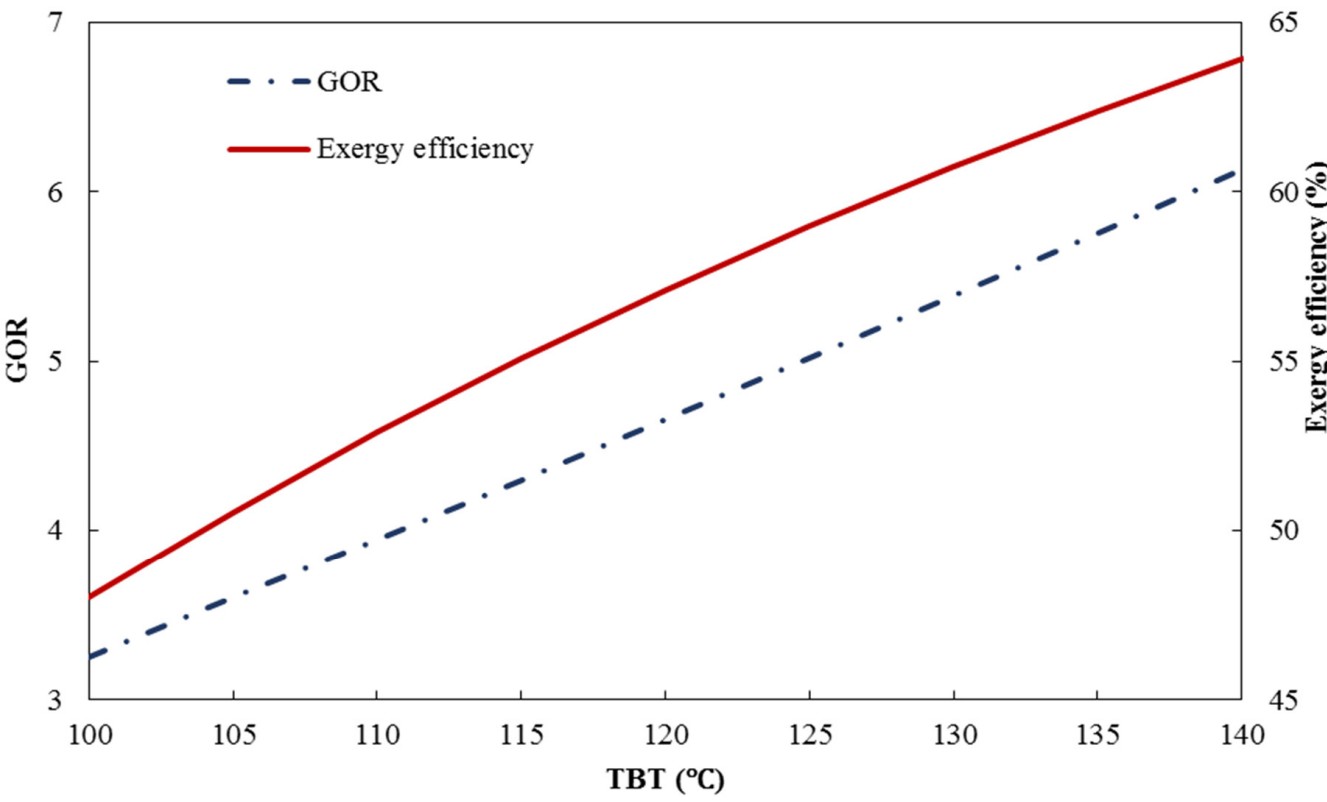

**Figure 8.** Variations of GOR and exergy efficiency in terms of top brine temperature.

The number of stages on gained output ratio (GOR) and exergy efficiency of the heat recovery section and heat rejection section are displayed in Figure 10A,B, respectively. As mentioned, the amount of distillate water production increases with increasing the number of stages which leads to an increase in GOR for a given heating steam flow. Moreover, with increasing the number of stages, the exergy efficiency of the heat recovery section continuously decreases while the exergy efficiency of the heat rejection section increases up to a maximum point at stage six, and then it decreases. The reason is that, due to a decrease in temperature in the last stages, the quality of distillate vapor decreases which results in a reduction of exergy efficiency. Hence, six stages are an optimal number of stages in the heat rejection section.

*5.3. Exergoeconomic Analysis Results*

The exergoeconomic analysis is performed on the MSF-BR desalination system of the Abadan refinery complex with 18 stages in the heat recovery section and four stages in the heat rejection system. The results obtained for exergoeconomic and cost parameters are presented in Table 7 for each stage of the studied desalination system. It is concluded that the total cost rate $\dot{Z}_{tot}$ increases from the first stage to the last stage due to an increase in heat transfer area and maintenance costs. In addition, an increase in stage number leads to an increase in the heat transfer rate of the brine heater which increases in fuel cost rate ($\dot{C}_f$). Moreover, the distillate water production increases which increases the product cost rate ($\dot{C}_p$). This is while increasing the stage number, the exergy destruction cost rate firstly increases and then it decreases.

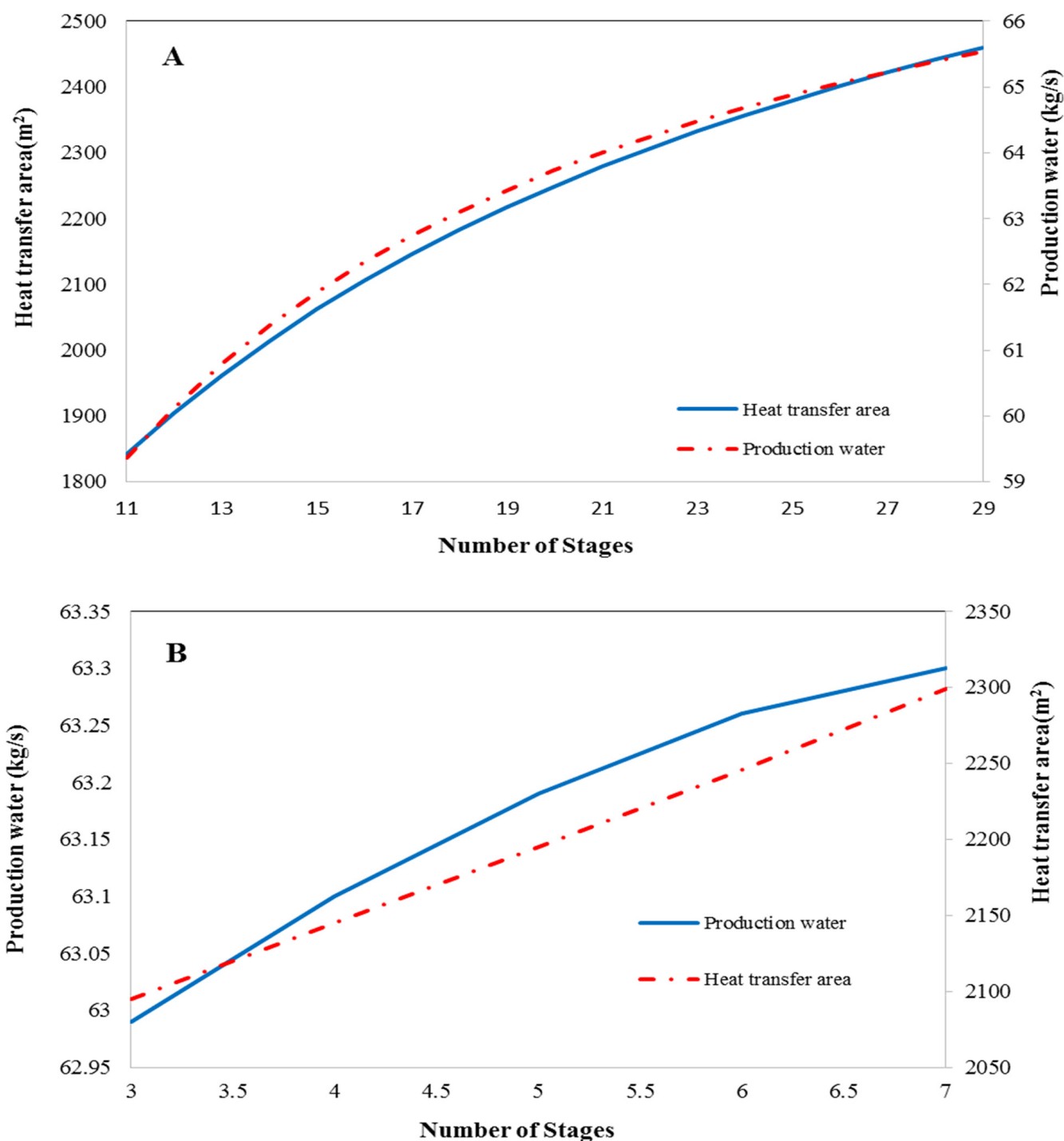

**Figure 9.** Variations of production water and heat transfer area in terms of the number of stages for (**A**) heat recovery section and (**B**) heat rejection section.

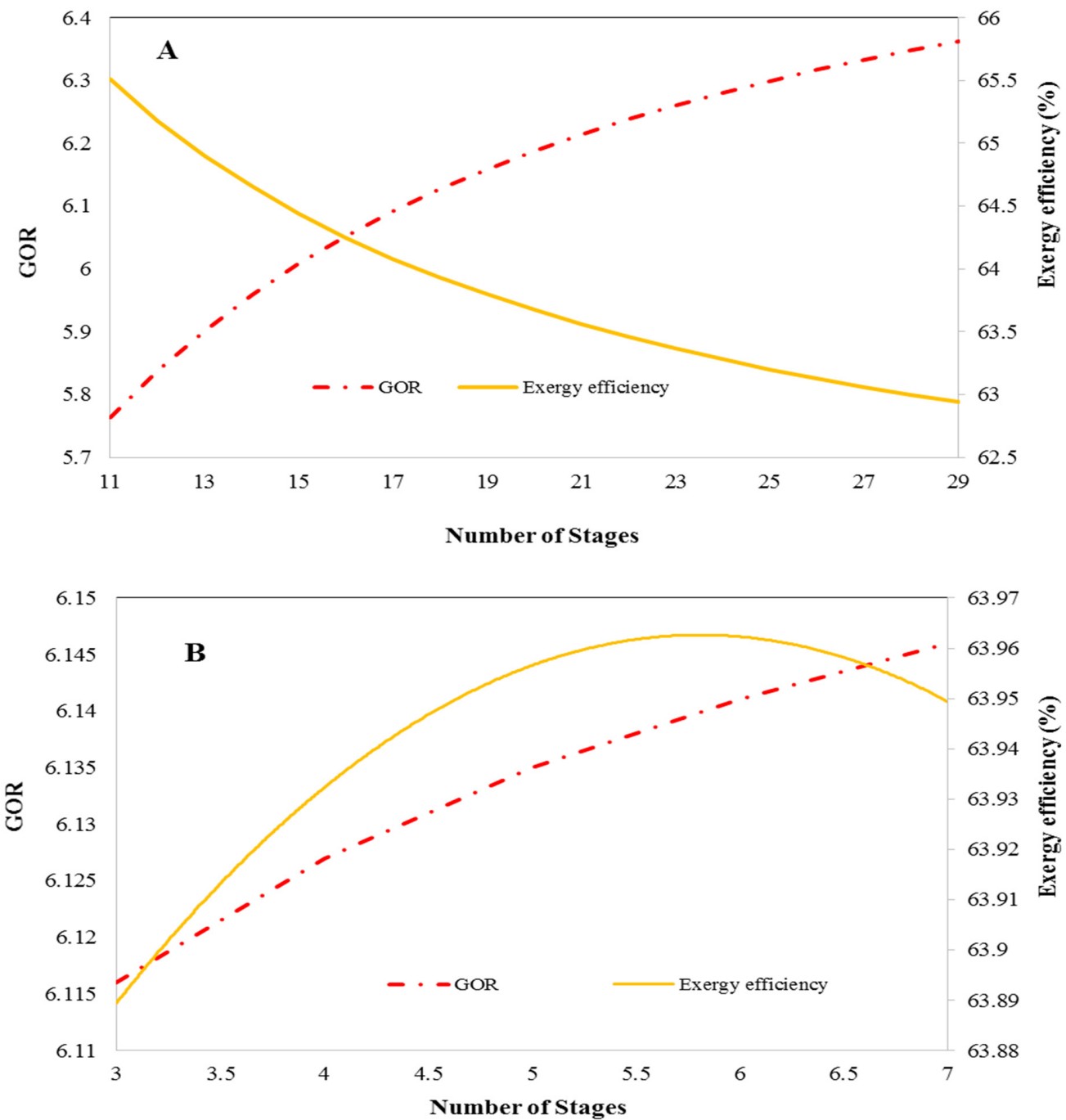

**Figure 10.** Variations of GOR and exergy efficiency in terms of the number of stages. (**A**) Heat recovery section and (**B**) heat rejection section.

As it was shown in Figure 10, with increasing the stages, the exergy efficiency increases at first and then decreases. It means that the exergy destruction firstly decreases and then increases.

The parameter of relative cost difference (r), which denotes the difference between product cost and fuel cost in a stage, increases with increasing the stage number. The exergoeconomic factor which is defined as the ratio of stage total cost rate to the sum of stage total cost rate and exergy destruction cost rate indicates the effect of the hidden cost compared with the total cost. Since the exergy destruction cost rate firstly increases and then decreases, therefore, the exergoeconomic factor decreases at first and then increases.

**Table 7.** Exergoeconomic and cost parameters for each stage of the MSF-BR desalination system.

| Stage | $\dot{C}_f(\frac{\$}{h})$ | $\dot{C}_p(\frac{\$}{h})$ | $\dot{C}_D(\frac{\$}{h})$ | $\dot{Z}_{total}(\frac{\$}{h})$ | f (%) | r |
|---|---|---|---|---|---|---|
| 1 | 18.13 | 19.78 | 7.29 | 0.60 | 7.6 | 0.09 |
| 2 | 21.15 | 36.58 | 9.09 | 0.54 | 5.6 | 0.73 |
| 3 | 26.75 | 59.53 | 14.64 | 0.52 | 3.4 | 2.9 |
| 4 | 37.32 | 81.95 | 18.49 | 0.50 | 2.63 | 1.19 |
| 5 | 48.08 | 106 | 21.44 | 0.49 | 2.23 | 1.2 |
| 6 | 59.24 | 131.2 | 23.56 | 0.48 | 2 | 1.2 |
| 7 | 70.8 | 157.6 | 24.86 | 0.49 | 2 | 1.22 |
| 8 | 82.79 | 185.3 | 25.34 | 0.49 | 1.9 | 1.24 |
| 9 | 95.22 | 214.5 | 24.98 | 0.50 | 2 | 1.25 |
| 10 | 108.1 | 245.4 | 23.8 | 0.51 | 2.1 | 1.27 |
| 11 | 121.5 | 278.1 | 21.82 | 0.52 | 2.1 | 1.28 |
| 12 | 135.5 | 313 | 19.05 | 0.53 | 2.7 | 1.3 |
| 13 | 150.1 | 350.3 | 15.72 | 0.55 | 3.4 | 1.33 |
| 14 | 165.3 | 390.5 | 11.29 | 0.57 | 4.8 | 1.63 |
| 15 | 181.4 | 434.1 | 6.45 | 0.60 | 8.5 | 1.39 |
| 16 | 198.5 | 481.7 | 5.77 | 0.63 | 9.8 | 1.43 |
| 17 | 216.8 | 533.9 | 3.21 | 0.66 | 17.5 | 1.46 |
| 18 | 240.3 | 600.7 | 1.93 | 0.87 | 31.1 | 1.5 |

Today, due to greenhouse emissions and associated costs, the efficiency improvement of energy systems that consume fossil fuels becomes a major concern. As described earlier, most thermal desalination systems use fossil fuels to desalinate seawater. The waste heat of refinery complexes can be a good source to supply the heating steam of thermal desalination systems. Table 8 presents the energy cost and emission penalty cost of the Abadan refinery complex. It is found that if the steam boiler of the MSF-BR desalination system is replaced by the waste heat of the refinery complex as heating steam, a significant amount of cost savings can be annually obtained.

**Table 8.** Energy consumption, emission production and associated costs in the Abadan refinery complex.

| Parameter | Value | Unit |
|---|---|---|
| Natural gas consumption | 34,272 | $m^3$/year |
| $CO_2$ emission production | $8 \times 10^7$ | kg/year |
| Natural gas cost | 9103 | \$/year |
| Emission penalty cost | $193 \times 10^4$ | \$/year |

## 6. Conclusions

In this study, the performance of a multi-stage flash brine recirculation (MSF-BR) system powered by the waste heat of the Abadan refinery complex as the heating steam is investigated based on energy, exergy and exergoeconomic analysis. Furthermore, the effects of top brine temperature (TBT), number of stages and ambient temperature on the performance of the desalination system was studied. The main conclusions are as follows:

- With increasing the top brine temperature, the exergy efficiency, gained output ratio (GOR) and distillate water production increase by 34%, 47% and 47% respectively.
- With increasing the number of stages in the heat recovery section, GOR and distillate water production increase while the exergy efficiency decreases.
- With increasing the number of stages in the heat rejection section, distillate water production increases while GOR firstly increases and then decreases. Thus, the optimal number of stages in the heat rejection section is six stages.
- The relative cost difference increases by 94% with increasing the stage number.
- With increasing the stage number, the exergoeconomic factor firstly decreases and then increases.
- By utilizing the waste heat of the refinery complex as heating steam, 9103 $/year energy cost savings and $193 \times 10^4$ $/year emission penalty cost savings are obtained.

**Author Contributions:** Conceptualization, M.D.-D. and E.T.; methodology, E.T.; software, F.F.; validation, M.D.-D., E.T. and F.F.; formal analysis, F.F.; investigation, M.D.-D.; resources, F.F.; data curation, F.F.; writing—original draft preparation, E.T.; writing—review and editing, M.D.-D.; visualization, F.F.; supervision, M.D.-D.; project administration, E.T.; funding acquisition, E.T. All authors have read and agreed to the published version of the manuscript.

**Funding:** This research received no external funding.

**Data Availability Statement:** Not applicable.

**Acknowledgments:** The authors would like to thank/ the Abadan refinery complex and Engineer Alizadeh for providing technical and financial support during this project.

**Conflicts of Interest:** The authors declare no conflict of interest.

## Nomenclature

| | | | |
|---|---|---|---|
| MSF | Multi-stage flash | Subscript | |
| MSF-BR | Multi-stage flash with brine recycle | P | Pump |
| MED | Multi-effect desalination | HEX | Heat exchanger efficiency |
| GOR | Gain output ratio | In | Input |
| RO | Reverse osmosis | Fg | Vaporization condition |
| ED | Electro dialyze | Sat | Saturation |
| T | Temperature (°C) | Out | Output |
| P | Pressure (kPa) | Br | Brine |
| Q | Thermal energy (kW) | Eq | Equilibrium |
| M | Mass flow rate (kg/s) | St | Stage |
| h | Enthalpy (kJ/kg) | s | Steam produce |
| A | Area (m$^2$) | ex | Exergy |
| Ex | Exergy (kW) | cv | Control volume |
| W | Work (kJ) | | |

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
