# Peer review of "Studying a Multi-Stage Flash Brine Recirculation (MSF-BR) System Based on Energy, Exergy and Exergoeconomic Analysis"

_water, doi:10.3390/w14193108_

Round 1

Reviewer 1 Report

This manuscript discussed about industry scale multi-stage flash brine recirculation system modeling study. It is an interesting work and I see large  potential to other water shortage countries & areas. But this work also has some parts which could be improved. I would suggest authors to address these questions below.

(1) There are some typos like "Codensate" ,"Tempreture", "steges" in different figures and main texts, please revise accordingly and check again.

(2) This manuscript has a lot of different fonts/font sizes/styles, I suggest it to be consistent.

(3) Abstract needs 1-2 sentences to talk about potential to the industry and others. Conclusion is OK with these bullet writings but I still suggest to have added some perspective not only technical results. What can other readers get from this work?

(4) Fig.1, does it need to introduce a little about other techniques? Some abbreviations even don't show up until after acknowledgement.

(5) Fig.2, good schematics and actually the most important figure. I suggest redraw, revise the typo, take out all words shadow,  draw it more clearer thinking about BFD or PFD to improve, also making point clearer.  1,2,3,.22 try letting readers know it is stage.

(6) Fig.3, can authors point out specific part in this chemical plant figure accordingly fig.2? No need to point out all. 

(7) The title is about energy, exergy and exergoeconomic, the authors should explain these in the introduction, please point out or revise.

Author Response

Reviewer #1:

This manuscript discussed about industry scale multi-stage flash brine recirculation system modeling study. It is an interesting work and I see large potential to other water shortage countries & areas. But this work also has some parts which could be improved. I would suggest authors to address these questions below.

Firstly, the authors would like to thank the reviewer for his helpful comments and also for the time spent on reviewing process. The detailed response to the comments is as follows:

(1) There are some typos like "Codensate" ,"Tempreture", "steges" in different figures and main texts, please revise accordingly and check again.

@Answer:

  • Thanks for the valuable reviewer’s comment. All sections of manuscript have been evaluated and all typos in the text and figures have been corrected as follows:

Figure 4. Variations of temperature and pressure in each stage.

Figure 6. Variations of distillate water in terms of ambient temperature for different number of stages.

Figure 7. Variations of distillate water and heat transfer area in terms of top brine temperature.

Figure 9. Variations of distillate water and heat transfer area in terms of number of stages for A) Heat recovery section and B) Heat rejection section.

(2) This manuscript has a lot of different fonts/font sizes/styles, I suggest it to be consistent.

@Answer:

Thanks for the valuable reviewer’s comment. The sizes and styles of the fonts have been modified in all sections of the revised version of manuscript.

(3) Abstract needs 1-2 sentences to talk about potential to the industry and others. Conclusion is OK with these bullet writings but I still suggest to have added some perspective not only technical results. What can other readers get from this work?

@Answer:

Thanks for the valuable reviewer’s comment. The Abstract sections have been modified in the revised version of manuscript as:

Abstract: Due to lack of natural water resources and high consumption of water in industries, desalination systems are good options to supply the water demands especially in regions with water crisis. On the other hand, there is a lot of heat waste in various industrial sectors. If these wastes are used in thermal desalination cycles, in addition to improving efficiency and reducing energy consumption, the production of environmental pollutants can also be reduced. In this paper, the multi-stage flash brine recirculation (MSF-BR) system of Abadan refinery is investigated from energy-exergy-exergoeconomic viewpoints. In addition, the effects of top brine temperature (TBT), number of stages and ambient temperature on performance of the system is evaluated. The results at maximum brine temperature show that with increasing the TBTthe exergy efficiency, gained output ratio (GOR) and distillate water production increase by 34%, 47% and 47% respectively. It is also found that if the number of stages in heat rejection section increases to more than 6 stages, GOR will decrease. The exergoeconomic analysis results reveal that the relative cost difference increases by 94% with an increase in the number of stages. Finally, it is concluded that by using the waste heat of refinery complex as heating steam to run the desalination system, 9103 $/year cost saving due energy consumption reduction and 193×104 $/year cost saving due CO2 emission reduction are obtained.

(4) Fig.1, does it need to introduce a little about other techniques? Some abbreviations even don’t show up until after acknowledgement.

@Answer:

Thanks for the valuable reviewer’s comment. Additional explanations regarding desalination methods and abbreviations related to Fig. 1 were added in the introduction section as follows:

The climate change and world population growth have led to increase in water demand and consequently decrease in water resources [1]. Around 40% of world’s population suffer from lack of potable water which is expected to be increased in future [2]. Although 70% of earth’s surface is covered with water but only 3% of it is drinking water [3]. Desalination of seawater is one of the growing methods to supply potable water around the world [4]. Generally, the desalination methods are divided into thermal and electrical techniques. Thermal desalination systems are generally divided into two main methods, MSF (multi stage flash) and MED (multi effect desalination). The water production process in thermal systems uses evaporation and distillation processes. Therefore, the produced water is of good quality. Power water desalination systems, the most common of which is the RO (reverse osmosis) method, use a membrane to filter and purify water. Also, Electrodialysis (ED) and nanofiltration (NF) are another desalination method using membrane. While employing thermal desalination methods leads to pollutant emissions due to using fossil fuels but the waste heat of industrial units can be utilized as a heat source for thermal desalination systems. As shown in Fig. 1, after reverse osmosis (RO), thermal desalination technologies account for a significant share of total desalination methods in the world. Therefore, optimization of such thermal desalination systems has considerable impact on reduction of fuel consumption and pollutant emissions. Multi-stage flash (MSF) is one of the most used thermal desalination techniques [5]. However, MSF desalination systems consume a large amount of fossil fuels which leads to increase in air pollution and decrease in non-renewable energy sources. As a result, energy consumption optimization of thermal desalination units like MSF results can reduce the energy consumption and water production cost [6].

(5) Fig.2, good schematics and actually the most important figure. I suggest redraw, revise the typo, take out all words shadow, draw it more clearer thinking about BFD or PFD to improve, also making point clearer.  1,2,3,.22 try letting readers know it is stage.

@Answer:

Thanks for the valuable reviewer’s comment. Fig. 2 has been corrected in the revised version of manuscript as follows:

Figure 2. Schematic view of MSF-BR unit.

(6) Fig.3, can authors point out specific part in this chemical plant figure accordingly fig.2? No need to point out all.

@Answer:

Fig. 3 is a real picture of the desalination system under consideration. Due to the complexity of the real cycle, it is not possible to provide complete details.

(7) The title is about energy, exergy and exergoeconomic, the authors should explain these in the introduction, please point out or revise.

@Answer:

More explanations regarding energy, exergy and economic exergy methods has been added in the introduction section of revised version of the manuscript as follows:

Iran is one of those countries that is experiencing some serious water crisis in the world. Industries such as chemicals, food, and metals account for a large part of water consumption. Therefore, the use of desalination plants is a good option to tackle the water shortage and supply the water demand. Furthermore, the waste heat of petroleum refineries located in coastal areas can be utilized for thermal desalination of seawater. This study investigates the multi-stage flash brine recirculation (MSF-BR) system of Abadan refinery which has 18 stages in heat recovery section and 4 stages in heat rejection section.

In the first stage, the current research has investigated and modeled the water desalination system installed in the refinery. The system in the refinery uses natural gas to supply the energy required for the desalination system. But according to field investigations, heat losses in the steam production sector are significant. Therefore, in this research, the feasibility of using existing steam waste with the aim of supplying the energy needed for the desalination system is proposed. The potential of employing waste heat of refinery complex as the heating steam of desalination unit is studied based on energy, exergy and exergoeconomic analysis. The main goal of choosing energy, exergy and exergy analysis is to achieve a logical optimization by considering important aspects. Exergoeconomics is the branch of thermodynamic that combines exergy and economic analysis to provide the system designer with information not available through conventional energy analysis and economic evaluations.

Reviewer 2 Report

Accept

Author Response

@Answer:

We would like to thank the respected reviewer for his detailed review of the article and his/her time. Also, we are very grateful for the good opinion of the respected reviewer regarding this research.

Reviewer 3 Report

The paper carries out an extensive investigation in terms of energy, exergy, and economic analysis. However, it fails to express why a research study on this topic is necessary or what are the new findings. What are the research gaps of this study or what are the objectives? To me, it feels like no new information is available from this investigation. Hence, I do not recommend its publication. 

Author Response

@Answer:

Many thanks to the respected referee for carefully reviewing the paper. More explanations about the importance and necessity of conducting research were added as follows:

Iran is one of those countries that is experiencing some serious water crisis in the world. Industries such as chemicals, food, and metals account for a large part of water consumption. Therefore, the use of desalination plants is a good option to tackle the water shortage and supply the water demand. Furthermore, the waste heat of petroleum refineries located in coastal areas can be utilized for thermal desalination of seawater. This study investigates the multi-stage flash brine recirculation (MSF-BR) system of Abadan refinery which has 18 stages in heat recovery section and 4 stages in heat rejection section.

In the first stage, the current research has investigated and modeled the water desalination system installed in the refinery. The system in the refinery uses natural gas to supply the energy required for the desalination system. But according to field investigations, heat losses in the steam production sector are significant. Therefore, in this research, the feasibility of using existing steam waste with the aim of supplying the energy needed for the desalination system is proposed. The potential of employing waste heat of refinery complex as the heating steam of desalination unit is studied based on energy, exergy and exergoeconomic analysis. The main goal of choosing energy, exergy and exergy analysis is to achieve a logical optimization by considering important aspects. Exergoeconomics is the branch of thermodynamic that combines exergy and economic analysis to provide the system designer with information not available through conventional energy analysis and economic evaluations.

Round 2

Reviewer 1 Report

The manuscript has revised carefully and answered all the questions. Herein I would recommend to publish in this form.

Reviewer 3 Report

I think the authors have provided clarification, and their work could well be useful for industrial researchers although the academic community might not find anything new. I recommend its publication for its overall summary and a one-stop paper for a comprehensive thermodynamic analysis.